# Diversity and antibiotic susceptibility profiles of bacterial isolates from wound infections in patients at the surgical unit of Kisii teaching and referral hospital, Kenya

**Samson Onsando**[1,2*], **Wycliffe Masanta**[3], **Andrew Nyerere**[2], **Moses Njire**[4], **Gervason Moriasi**[5,6]

1 Kisii Teaching and Referral Hospital, Kisii, Kenya, 2 Department of Medical Microbiology, College of Health Sciences, Jomo Kenyatta University of Agriculture and Technology, Nairobi, Kenya, 3 Department of Medical Microbiology, School of Medicine, Maseno University, Private Bag, Maseno, Kenya, 4 Department of Botany, Jomo Kenyatta University of Agriculture and Technology, Nairobi, Kenya, 5 Department of Medical Biochemistry, Medical School, Mount Kenya University, Thika, Kenya, 6 Department of Biochemistry, Microbiology, and Biotechnology, School of Pure and Applied Sciences, Kenyatta University, GPO, Nairobi, Kenya

* samonsando@gmail.com

## Abstract

Antimicrobial resistance (AMR) poses a growing threat to global health, with wound infections serving as significant reservoirs for multidrug-resistant bacteria. Understanding the socio-demographic, clinical, and microbiological characteristics of wound infections is critical for effective intervention strategies in resource-limited settings. Thus, this study aimed to evaluate the diversity, bacterial profiles, and resistance patterns of wound infections at Kisii Teaching and Referral Hospital. To do this, a total of 118 bacterial isolates from wound swabs were collected from 149 patients who met the eligibility criteria, from May to November 2024. These samples were cultured and analysed using standard microbiological techniques and antibiotic susceptibility testing according to the Clinical and Laboratory Standards Institute guidelines. Resistance genes associated with Vancomycin resistance in *Staphylococcus aureus* and *Staphylococcus xylosus* were also investigated using polymerase chain reaction (PCR) with appropriate primers and agarose gel electrophoresis. The results revealed a predominance of male patients (n = 79; 53.02%), avulsions (n = 50; 34.00%), and wounds on lower limbs (n = 63; 42.29%). *Escherichia coli* (n = 33; 27.97%), *Enterobacter gergoviae* (n = 20; 16.95%), *Staphylococcus aureus* (n = 18; 15.25%), and *Staphylococcus xylosus* (n = 18; 15.25%) were the most frequently isolated pathogens. Alarmingly high resistance rates were observed against ACCESS antibiotics, including Cotrimoxazole-Trimethoprim (n = 114; 97%), Ampicillin (n = 109; 92%), and Streptomycin (n = 100; 85%). Resistance to Watch and Reserve antibiotics, including Vancomycin (n = 11; 29%) and Ceftriaxone (n = 27; 33%), was also documented, particularly among Gram-positive isolates such as *Staphylococcus aureus*

**Data availability statement:** All data are in the manuscript and/or Supporting Information files.

**Funding:** The author(s) received no specific funding for this work.

**Competing interests:** The authors have declared that no competing interests exist.

and Gram-negative pathogens like *Pseudomonas aeruginosa*. Resistance genes (*Van*A, *Van*B, *Van*C) were detected in *S. aureus* and *S. xylosus*, underscoring the role of genetic mechanisms in resistance propagation. These findings highlight the critical need for targeted antimicrobial stewardship programmes and strengthened infection prevention measures in high-burden hospital settings, such as Kisii Teaching and Referral Hospital. This study underscores the importance of local resistance data in guiding empirical therapy and informs policies to mitigate AMR in resource-limited contexts. Moreover, enhanced surveillance and research into alternative therapies are crucial to safeguarding the efficacy of life-saving antibiotics and improving patient care outcomes.

## Introduction

Wound infections pose a significant global health challenge, particularly in resource-limited settings where healthcare infrastructure may be inadequate [1]. Recent estimates suggest that approximately 7% of hospitalised patients in low- and middle-income countries develop wound infections, with even higher rates in surgical wards [2]. The diversity of bacterial pathogens associated with wound infections is extensive and dynamic, influenced by factors such as the nature of the wound, the patient's immune status, and the surrounding environment [3]. These infections can delay wound healing, increase morbidity, and, in severe cases, lead to systemic conditions such as sepsis [4]. Effective management of wound infections requires a thorough understanding of the bacterial flora involved and their antibiotic sensitivity profiles [3]. Without such data, clinicians face challenges in selecting effective empirical therapies.

The surgical unit, as a critical healthcare domain, frequently manages wound infections stemming from trauma, surgical procedures, burns, and chronic conditions such as diabetic foot ulcers [5]. Globally, trauma-related wound infections account for nearly 30% of admissions in surgical units and are often associated with higher incidences of morbidity and debility [6]. Infections in such settings are often polymicrobial, involving Gram-positive, Gram-negative, and occasionally anaerobic bacteria, including *Staphylococcus aureus*, *Pseudomonas aeruginosa, Escherichia coli, Klebsiella* spp., and *Proteus spp.* among others [7]. However, the prevalence and diversity of these pathogens can vary geographically and temporally, necessitating localised studies to inform evidence-based interventions [8,9]. Furthermore, environmental factors and hospital infection control practices significantly influence the microbial landscape, making continuous surveillance critical [10].

The rising prevalence of antimicrobial resistance (AMR) further complicates the management of wound infections [11]. According to the World Health Organisation, over 4.71 million deaths annually are attributed to bacterial infections, with about 1.14 million deaths attributed to bacterial AMR, with a substantial proportion linked to bacterial infections [12]. Misuse and overuse of antibiotics in clinical settings have accelerated the emergence of resistant strains, including methicillin-resistant

*Staphylococcus aureus* (MRSA), extended-spectrum beta-lactamase (ESBL)-producing Enterobacteriaceae, and carbapenem-resistant *Pseudomonas* and *Acinetobacter* species [13]. This resistance compromises therapeutic outcomes, leading to prolonged hospital stays, increased healthcare costs, and higher mortality rates [11]. Alarmingly, studies report resistance rates exceeding 60% for some commonly used antibiotics in sub-Saharan Africa, underscoring the urgency of evidence-based and targeted interventions [12,14]. Previous studies in Kenya have highlighted the presence of multidrug-resistant bacteria in various clinical samples [15–17], but data specific to wound infections in this region remain scarce, hindering the development of tailored antimicrobial stewardship strategies and effective infection control measures.

Therefore, this study aimed at characterising the diversity of bacterial isolates from wound infections in patients attending the surgical unit at Kisii Teaching and Referral Hospital (KTRH). Additionally, it evaluated the antibiotic sensitivity profiles of these isolates to guide empirical treatment strategies and inform antimicrobial stewardship programmes. This study's findings will contribute to improved patient outcomes and reduce the burden of wound infections in within the region, and beyond. By providing a robust evidence base, this research can also support policy development and resource allocation for managing AMR in similar healthcare settings.

## Methodology

### Study site

This study was conducted at Kisii Teaching and Referral Hospital (KTRH), located in Southwestern Kenya (0°40'15.2"S, 34°46'17.0"E). As a major referral centre in the Nyanza province, KTRH serves over 6 million people, including residents from South Nyanza, the Rift Valley, and northern parts of Tanzania. The hospital offers specialised services, including a well-equipped laboratory capable of microbiological analyses such as bacterial culture, identification, and antimicrobial susceptibility testing.

This study focused on the Surgical Unit, which manages a high volume of diverse wound types, including traumatic wounds (e.g., from road traffic accidents and agricultural injuries), surgical wounds, and chronic wounds (e.g., diabetic foot ulcers and pressure ulcers). Common sources of wounds include road traffic accidents, agricultural activities, and chronic conditions like diabetes. During the six-month study period (May–November 2024), the unit attended to an average of 300–350 patients monthly, providing a robust sample size for investigating bacterial susceptibility and resistance patterns.

KTRH has an Antimicrobial Stewardship Programme (ASP) and Infection Prevention and Control (IPC) policy to promote rational antibiotic use, monitor resistance, and minimise healthcare-associated infections. The unit's role as a referral centre for complex cases, including multidrug-resistant infections, and its diverse patient population—many with limited access to primary healthcare—made it an ideal setting for studying wound infections and antimicrobial resistance in resource-limited contexts.

### Study design

This study adopted a descriptive cross-sectional design to characterise the diversity of bacterial isolates and determine their antibiotic sensitivity profiles.

### Study population and sample size

The study targeted patients attending the surgical unit at KTRH with clinical signs of wound infections, including purulent discharge, erythema, swelling, or delayed wound healing. Considering the Surgical Unit at KTRH attends to an average of 300–350 patients per month with various wound types, over the six-month study period (May–November 2024), this translates to a total population of approximately 1,950 patients. Besides, since 60% of the patients met the inclusion criteria, the eligible population was estimated to be: $1,950 \times 0.60 = 1,170$ eligible patients.

The formula described previously [18] (Eq. 1), as adapted by [19] was used to determine the ideal sample size for this study.

$$n = \frac{Z^2 p(1-p)}{d^2}$$

(1)

Where n = sample size; Z = the Z-score for a 95% confidence level (1.96); p = 0.25, which was the estimated prevalence of antimicrobial resistance among patients with wound infections in Eastern African hospitals [20]; d = p)recision or margin of error (0.05).

Therefore,

$$n = \frac{1.96^2 \times 0.25(1-0.25)}{0.05^2}$$

$$n = 288$$

Adjusting for the finite population (N = 1,170):

$$n_{adjusted} = \frac{n}{1 + \left(\frac{n-1}{N}\right)}$$

$$n_{adjusted} = \frac{288}{1 + \left(\frac{288-1}{1,170}\right)}$$

$$n_{adjusted} = 231.27 = 231 \text{ patients}$$

## Inclusion and exclusion criteria

Patients were considered eligible for inclusion in the study if they presented with clinically diagnosed infected surgical wounds, as confirmed by clinical evaluation (e.g., presence of purulent discharge, erythema, warmth, swelling, or pain), had not received any antibiotic treatment within the preceding 48 hours, and provided informed consent to participate in the study.

Conversely, patients with systemic infections unrelated to their surgical wounds, and those on prolonged immunosuppressive therapy (e.g., corticosteroids, biologics, or chemotherapy) were excluded from this study.

## Collection of demographic data and clinical characteristics of study patients

The demographic information of the recruited patients including their gender and age, as well as relevant clinical data including the type of wound and anatomic location of the wound were collected using a standard clinical form designed by KUTRH.

## Wound site preparation and sample collection

Wounds were initially rinsed with sterile normal saline to minimise contamination as recommended previously [21]. For wounds covered with dressings, the dressing was carefully removed, and the area was thoroughly cleansed with sterile normal saline to eliminate surface exudates, necrotic tissue, and skin flora. The samples, including pus, purulent exudates, and other secretions, were collected aseptically using sterile cotton swabs (Deltalab) [22]. Adequate pressure was applied during swabbing to ensure sufficient specimen retrieval from the wound bed, without causing further injury or pain to the patient. All specimens were transported to the KTRH Microbiology Laboratory within 30 minutes, logged by laboratory personnel involved in the study and immediately processed for microbial culture and identification.

## Microbiological culturing and identification procedure

The collected samples were cultured on Sheep Blood Agar (BA), MacConkey Agar (MCA), Ashdown Agar (ADA), and Cystine Lactose Electrolyte Deficient (CLED) media, prepared according to standard protocols [23,24]. Briefly, commercially available powdered media were reconstituted with triple-distilled water and sterilised by autoclaving at 121°C for 15 minutes, following the manufacturer's guidelines. Inoculation onto culture media was performed using the streak plate method, where a sterile wire loop was aseptically used to streak the specimens, to ensure the formation of discrete colonies. The inoculated plates were incubated under both aerobic and anaerobic conditions at 37°C for 24 hours, with extended incubation of up to five days for Ashdown Agar to allow the growth of slow-growing organisms. Subculturing under the same conditions was undertaken to obtain pure colonies. The most predominant colonies were isolated using sterile swabs for further characterisation. Besides, colony morphology was recorded, and a series of biochemical and phenotypic including Gram staining, oxidase, catalase, and coagulase tests, as well as the API test systems (API 20 E, API 20 NE, and API STAPH) were performed for precise microbial identification. Further confirmation of bacterial species was performed using an automated VITEK® 2 compact system [25]. The isolates were preserved at −20°C for downstream applications, including antimicrobial susceptibility testing, DNA extraction, and polymerase chain reaction (PCR) analysis.

## Antibiotic susceptibility testing

The antibiotic susceptibility of bacterial isolates was evaluated using the Kirby-Bauer disk diffusion assay, according to Clinical and Laboratory Standards Institute (CLSI) guidelines [26]. The bacterial isolates were inoculated onto Mueller-Hinton agar plates using a standard procedure to achieve uniform distribution. Antibiotic disks (Octo Discs) including Ampicillin (25 µg), Tetracycline (100 µg), Co-trimoxazole (Sulphathiazole/Trimethoprim) (23.75/1.25 µg), Nitrofurantoin (200 µg), Nalidixic acid (30 µg), Streptomycin (25 µg), and Gentamicin (10 µg), Vancomycin(30 µg), Ceftriaxone(30 µg), Linezolid(30 µg), and Ofloxacin (5 µg), were carefully placed on the agar surface, and plates were incubated under optimal conditions to promote bacterial growth and inhibition. Following incubation, the diameters of the inhibition zones were measured precisely using a calibrated ruler, recorded, interpreted according to CLSI breakpoints, and categorised as susceptible, intermediate, or resistant [23,26,27].

## DNA extraction, quantification, and analysis

**DNA extraction.** Bacterial DNA was extracted using the Quick-DNA™ Miniprep Kit (Zymo Research) according to the manufacturer's protocol [28]. Briefly, 500 µL of Genomic Lysis Buffer was added to the cell pellet, followed by vortexing for 5 seconds to lyse the cells. The mixture was incubated at room temperature for 10 minutes before being transferred to a Zymo-Spin™ IICR column placed in a collection tube. The assembly was centrifuged at 10,000 rpm for 1 minute, and the flow-through was discarded. After that, the Zymo-Spin™ IICR column was transferred to a fresh collection tube, and 200 µL of DNA Pre-Wash Buffer was added. The tube was centrifuged at 10,000 rpm for 1 minute, and the flow-through was discarded. Subsequently, 500 µL of DNA Wash Buffer was added to the column and centrifuged at the same speed for 1 minute. Then, the column was transferred to a clean microcentrifuge tube, and 50 µL of DNA Elution Buffer was added directly onto the column matrix. Following a 5-minute incubation at room temperature, the tube was centrifuged at 8,000 rpm for 1 minute to elute the DNA. The spin column was discarded, and the eluted DNA was stored at −20°C.

## DNA quantification and purity assessment

The DNA concentration was measured spectrophotometrically at 260 nm. Briefly, 495 µL of 10 mM Tris-Cl (pH 7.0) buffer was transferred to two cuvettes (500 µL capacity) and used to blank the spectrophotometer. To the second cuvette, 5 µL of DNA sample was added, thoroughly mixed with the buffer, and its absorbance at 260 nm ($A_{260}$) was recorded and used to calculate its concentration as follows:

$$\text{DNA concentration } (\mu g/ml) = \text{spectrophotometric conversion} \times (A_{260} - \text{correction factor}) \times \text{dilution factor}$$

Where spectrophotometric conversion = 50 µg/mL for dsDNA, correction factor = 0.012, and dilution factor = 100.

Samples with concentrations ≥50 ng/µL were used for further analysis.

Moreover, the purity of DNA was assessed by measuring absorbance at 280 nm (A280). The ratio $\frac{A_{260}}{A_{280}}$ was calculated to determine purity, with values ≥1.80 considered acceptable for PCR applications.

## Polymerase Chain Reaction (PCR)

PCR amplification was performed using the SimpliAmp™ Thermal Cycler (Applied Biosystems, Thermo Fisher Scientific, Germany). The reaction volume was set to 25 µL and consisted of 10 µM Forward Primer (0.5 µL), 10 M Reverse Primer (0.5 µL), template DNA (2.0 µL), Taq 2X Master Mix (12.5 µL), and Nuclease-free water (9.5 µL). Primers targeting specific resistance genes (*vanA*, *vanB*, and *VanC*) were used to amplify products of defined sizes, as shown in Table 1. Annealing temperatures for each primer pair were optimised (54°C for *vanA*, *vanB*, and *vanC*). The PCR conditions were as summarised in Table 2:

## Gel electrophoresis

DNA amplicons were resolved using 1% (w/v) agarose gel electrophoresis. To prepare the gel, 0.5 g of agarose was dissolved in 50 mL of 1X TAE buffer by heating in a microwave. After cooling to approximately 50°C, 2 µL of ethidium bromide was added for nucleic acid staining. The gel was poured into a tray fitted with an 8-well comb and allowed to set for 30 minutes. The DNA samples were prepared by mixing 10 µL of each PCR product with 2 µL of loading dye. A 5 µL DNA ladder (New England Biolabs) was loaded into the first well as a size marker. The gel was run at 120 V for 30 minutes using a Cleaver Scientific electrophoresis system (UK). Images were captured and documented using the gelLITE Gel Documentation System (Cleaver Scientific, UK).

## Data analysis

The obtained data were tabulated and organised in Microsoft Excel (Microsoft 365) and then exported to GraphPad Prism statistical software (version 10.3) for Mac for analysis (GraphPad Software, Boston, Massachusetts USA, www.graphpad.com)

**Table 1. Target genes and their respective primers.**

| Target gene | Primer Sequence (5' to 3') | | Product size |
|---|---|---|---|
| *van A* | F | CATGAATAGAATAAAAGTTGCAATA | 1000 bp-1050 bp |
| | R | CCCCTTTAACGCTAATACGATCAA | |
| *van B* | F | GTGACAAACCGGAGGCGAGGA | 350 - 400 bp |
| | R | CCGCCATCCTCCTGCAAAAAA | |
| *van C* | F | GAAAGACAACAGGAAGACCGC | 700 - 750 bp |
| | R | ATCGCATCACAAGCACCAATC | |

**Table 2. PCR conditions.**

| Step | Temperature | Time | Cycles |
|---|---|---|---|
| Initial Denaturation | 94°C | 5 minutes | 1 |
| Denaturation | 94°C | 30 seconds | 35 |
| Annealing | 54°C | 30 seconds | 35 |
| Extension | 72°C | 30 seconds | 35 |
| Final Extension | 72°C | 5 minutes | 1 |
| Hold | 4°C | Infinite | – |

[29]. Descriptive statistics were performed, and the resulting frequencies and percentages were used to summarise the patient demographics and diversity of bacterial isolates. Antibiotic resistance patterns were examined according to the CLSI guidelines [23,26,27] and tabulated.

## Ethical considerations

Ethical approval was obtained from the KTRH Ethics and Research Committee, University of Eastern Africa Baraton (REC: UEAB/ISERC/08/05/2024). Written informed consent was obtained from all adult participants. For patients aged below 18 years, their parents or legal guardians provided informed consent according to ethical research guidelines. Confidentiality of patient information was maintained throughout the study, and all procedures adhered to the Declaration of Helsinki principles.

## Results

### Demographic and clinical characteristics of study patients

Out of 221 patients, 149 consented to and participated in this study, representing a 67.42% response rate. This response rate was deemed statistically significant for detecting trends in bacterial diversity, susceptibility, and resistance patterns, given the prevalence of wound infections reported in a similar study setting [30]. The results showed that most patients from whom bacterial isolates were obtained were male (n = 79; 53.02%) (Table 3; S1_File). It was observed that most of the patients were aged between 31 and 40 years (n = 33; 22.15%). Besides, Avulsions constituted most of the wound types accounting for 34% (n = 50), while burns were the least frequent wounds (n = 7; 5%) in the patients (Table 3; S1_File). Moreover, the lower limb wounds were the most common (n = 63; 42.29%) compared to upper limb wounds which accounted for about one third (n = 50; 33.56%) (Table 3; S1_File).

**Table 3. Socio-demographic and clinical characteristics of the 149 study participants.**

| Characteristics | Categories | Frequency (n) | % |
|---|---|---|---|
| Sex | Female | 70 | 46.98 |
| | Male | 79 | 53.02 |
| Age (Years) | 0-10 | 9 | 6.04 |
| | 11-20 | 19 | 12.75 |
| | 21-30 | 22 | 14.77 |
| | 31-40 | 33 | 22.15 |
| | 41-50 | 18 | 12.08 |
| | 51-60 | 21 | 14.09 |
| | 61-70 | 11 | 7.38 |
| | 71-80 | 16 | 10.74 |
| Wound type | Avulsions | 50 | 34.00 |
| | Abrasion | 37 | 25.00 |
| | Laceration | 32 | 21.00 |
| | Burns | 7 | 5.00 |
| | Punctures | 23 | 15.00 |
| Anatomic site | Trunk | 36 | 24.16 |
| | Upper Limb | 50 | 33.56 |
| | Lower Limb | 63 | 42.29 |

## Bacterial strains isolated from wound swab cultures

In this study, ab total of 118 bacterial strains were isolated from the patient wounds. It was observed that most of the bacterial isolates from the wound swabs included *E. coli* (n = 33; 27.97%), *E. gergoviae* (n = 20; 16.95%), and *S. aureus* and *S. xylosus* (n = 18; 15.25%) (Table 4; S1_File). Conversely, the least frequent isolates included *S. intermedius, E. aerogenes, P. vulgaris, P. mirabillis*, and *P. haemolytica*, accounting for a paltry 0.85% (n = 1) (Table 4; S1_File).

## Resistance of the bacterial isolates to various antibiotics

**Resistance to ACCESS antibiotics.** This study assessed the antibiotic resistance of 118 bacterial isolates against the ACCESS category of antibiotics including ampicillin, tetracycline, nitrofurantoin, nalidixic acid, streptomycin, cotrimoxazole-trimethoprim, and gentamycin, commonly used to treat wound infections [31], following CLSI guidelines [26]. Overall, the results revealed a considerably higher resistance of the isolated bacteria to Cotrimoxazole-Trimethoprim (n = 114; 97%) and Ampicillin (n = 109; 92%), with significant resistance observed to Nitrofurantoin (n = 85; 72%), Streptomycin (n = 100; 85%), and Nalidixic Acid (n = 80; 68%) (Table 5). Lower resistance rates were noted for Tetracycline (n = 66; 56%) and Gentamicin (n = 58; 49%) (Table 5; S1_File).

Notably, as the results in Table 5 indicate, *E. gergoviae* (n = 20) exhibited widespread resistance, including 100% to Streptomycin and Cotrimoxazole-Trimethoprim, and 95% to Ampicillin (n = 19). Similarly, *E. coli* (n = 33) displayed high resistance to Cotrimoxazole-Trimethoprim (97%) and Ampicillin (93.9%), with moderate resistance to Gentamicin (57.6%) (Table 5). Besides, *S. aureus* (n = 18) showed notable resistance to Cotrimoxazole-Trimethoprim (88.9%) and Streptomycin (66.7%), while *S. xylosus* (n = 18) exhibited 100% resistance to Cotrimoxazole-Trimethoprim and 77.8% to Nalidixic Acid as shown in Table 5 (S1_File).

Other less frequent isolates such as *P. vulgaris* (n = 1)*, P. mirabilis* (n = 1)*, Y. enterocolitica* (n = 2), and *B. pseudomallei* (n = 3) uniformly resisted multiple antibiotics (100%), highlighting their multidrug resistance. Rare strains like *E. sakazakii* (n = 2)*, A. hydrophila* (n = 2), and *P. aeruginosa* (n = 2), also showed higher resistance to multiple ACCESS antibiotics as shown in Table 5 (S1_File).

**Table 4. Bacterial strains (n = 118) isolated from wound swab cultures.**

| Bacterial Isolate | Frequency (n) | Percentage (%) |
|---|---|---|
| *Enterobacter gergoviae* | 20 | 16.95 |
| *Staphylococcus xylosus* | 18 | 15.25 |
| *Staphylococcus aureus* | 18 | 15.25 |
| *Staphylococcus intermedius* | 1 | 0.85 |
| *Escherichia coli* | 33 | 27.97 |
| *Serratia liquefaciens* | 5 | 4.24 |
| *Serratia odorifera 1* | 4 | 3.39 |
| *Enterobacter sakazaldi* | 2 | 1.69 |
| *Enterobacter aerogenes* | 1 | 0.85 |
| *Enterobacter cloacae* | 2 | 1.69 |
| *Proteus vulgaris* | 1 | 0.85 |
| *Proteus mirabillis* | 1 | 0.85 |
| *klebsiella ornithinofytica* | 2 | 1.69 |
| *Pausteurella haemolytica* | 1 | 0.85 |
| *Aeromonas hydrophila 1* | 2 | 1.69 |
| *Yersinia enterocolihca* | 2 | 1.69 |
| *Burkholderia pseudomallei* | 3 | 2.54 |
| *Pseudomonas aeruginosa* | 2 | 1.69 |

**Table 5. Frequency of resistance to ACESS Antibiotics by the bacterial isolates (n = 118).**

| Bacterial Isolate | n | Proportion of resistant bacterial isolates to the ACCESS category of antibiotics | | | | | | |
|---|---|---|---|---|---|---|---|---|
| | | AMP | TETRA | NITRO | NDA | STREP | COT | GENTA |
| *Enterobacter gergoviae* | 20 | 19 (95.0%) | 14 (70.0%) | 16 (80.0%) | 12 (60.0%) | 20 (100.0%) | 20 (100.0%) | 12 (60.0%) |
| *Staphylococcus xylosus* | 18 | 16 (88.9%) | 8 (44.4%) | 13 (72.2%) | 14 (77.8%) | 13 (72.2%) | 18 (100.0%) | 7 (38.9%) |
| *Staphylococcus aureus* | 18 | 14 (77.8%) | 7 (38.9%) | 9 (50.0%) | 11 (61.1%) | 12 (66.7%) | 16 (88.9%) | 6 (33.3%) |
| *Staphylococcus intermedius* | 1 | 1 (100.0%) | 0 (0.0%) | 0 (0.0%) | 1 (100.0%) | 0 (0.0%) | 1 (100.0%) | 0 (0.0%) |
| *Escherichia coli* | 33 | 31(93.9%) | 22 (66.7%) | 23 (69.7%) | 23 (69.7%) | 29 (87.9%) | 32 (97.0%) | 19 (57.6%) |
| *Serratia liquefaciens* | 5 | 5 (100.0%) | 2 (40.0%) | 3 (60.0%) | 4 (80.0%) | 4 (80.0%) | 5 (100.0%) | 2 (40.0%) |
| *Serratia odorifera 1* | 4 | 4 (100.0%) | 2 (50.0%) | 2 (50.0%) | 2 (50.0%) | 4 (100.0%) | 4 (100.0%) | 1 (25.0%) |
| *Enterobacter sakazaldi* | 2 | 2 (100.0%) | 2 (100.0%) | 2 (100.0%) | 1 (50.0%) | 2 (100.0%) | 1 (50.0%) | 1 (50.0%) |
| *Enterobacter aerogenes* | 1 | 1 (100.0%) | 1 (100.0%) | 1 (100.0%) | 0 (0.0%) | 1(100.0%) | 1 (100.0%) | 0 (0.0%) |
| *Enterobacter cloacae* | 2 | 2 (100.0%) | 0 (0.0%) | 2 (100.0%) | 1 (50.0%) | 1(50.0%) | 2 (100.0%) | 1 (50.0%) |
| *Proteus vulgaris* | 1 | 1 (100.0%) | 1 (100.0%) | 1 (100.0%) | 1 (100.0%) | 1(100.0%) | 1 (100.0%) | 1 (100.0%) |
| *Proteus mirabillis* | 1 | 1 (100.0%) | 1 (100.0%) | 1 (100.0%) | 1 (100.0%) | 1(100.0%) | 1 (100.0%) | 1 (100.0%) |
| *klebsiella ornithinofytica* | 2 | 2 (100.0%) | 1 (50.0%) | 2 (100.0%) | 2 (100.0%) | 2(100.0%) | 2 (100.0%) | 1 (50.0%) |
| *Pausteurella haemolytica* | 1 | 1 (100.0%) | 0 (0.0%) | 1 (100.0%) | 1 (100.0%) | 1 (100.0%) | 1 (100.0%) | 1 (100.0%) |
| *Aeromonas hydrophila 1* | 2 | 2 (100.0%) | 0 (0.0%) | 2 (100.0%) | 1 (50.0%) | 2 (100.0%) | 2 (100.0%) | 0 (0.0%) |
| *Yersinia enterocolihca* | 2 | 2 (100.0%) | 2 (100.0%) | 2 (100.0%) | 2 (100.0%) | 2 (100.0%) | 2 (100.0%) | 2 (100.0%) |
| *Burkholderia pseudomallei* | 3 | 3 (100.0%) | 2 (67.0%) | 3 (100.0%) | 2 (67.0%) | 3 (100.0%) | 3 (100.0%) | 2 (67.0%) |
| *Pseudomonas aeruginosa* | 2 | 2 (100.0%) | 1 (50.0%) | 2 (100.0%) | 1 (50.0%) | 2 (100.0%) | 2 (100.0%) | 1 (50.0%) |

The values in parenthesis represent the percentage of bacterial isolates which showed resistance to respective antibiotics based on the CLSI guidelines [32]. AMP: Ampicillin; TETRA: Tetracycline; NITRO: Nitrofurantoin; NDA: Nalidixic acid; STREP: Streptomycin; COT: Cotrimoxazole-Trimethoprim; GENTA: Gentamycin.

## Resistance to watch and reserve antibiotics

This study also evaluated resistance patterns in the 118 bacterial isolates against Watch and Reserve antibiotics, including Ofloxacin, Ceftriaxone, Vancomycin, and Linezolid [31]. Resistance to these critical antibiotics was observed across both Gram-positive and Gram-negative isolates, underscoring the growing threat of antimicrobial resistance, and possible therapeutic dead end. Among Gram-positive isolates (n = 37), resistance to Ofloxacin (n = 7), Ceftriaxone (n = 12), Vancomycin (n = 11), and Linezolid (n = 9) was 18%, 32%, 29%, and 24%, respectively (Table 6). Notably, *S. aureus* (n = 18) exhibited moderate resistance to Vancomycin (33%) and Ceftriaxone (27%), while *S. xylosus* (n = 18) demonstrated similar resistance rates to Ceftriaxone (33%) and Vancomycin (26%) (Table 6; S1_File). However, *S. intermedius* (n = 1) showed complete resistance to Ceftriaxone (100%), though it was susceptible to other antibiotics (Table 6; S1_File).

For the Gram-negative isolates (n = 81), the overall resistance rates to Ofloxacin and Ceftriaxone were 29% (n = 24), and 33% (n = 27), respectively (Table 6; S1_File). In this study, *E. coli* (n = 33) exhibited moderate resistance to Ofloxacin (18%) and Ceftriaxone (24%) (Table 6). Higher resistance rates to Ofloxacin and Ceftriaxone were observed in *E. gergoviae* (n = 8; 40% and n = 7; 35%), *S. liquefaciens* (n = 2; 40% and n = 3; 60%), and *B. pseudomallei* (n = 2; 66% and n = 1; 33%), respectively as shown in Table 6 (S1_File). Alarming levels of resistance by *P. aeruginosa* to Ofloxacin (n = 1; 50%) and Ceftriaxone (n = 2; 100%) were observed in this study. Similarly, *E. sakazakii* exhibited resistance to Ofloxacin (n = 1; 50%) and Ceftriaxone (n = 2; 100%) (Table 6; S1_File)). Moreover, *E. cloacae* displayed complete resistance to Ofloxacin (n = 1; 50%) and moderate resistance to Ceftriaxone (n = 1; 100%) (Table 6; S1_File), while the other isolates, such as *P. vulgaris* and *P. haemolytica* were susceptible (Table 6; S1_File).

**Table 6. Proportion of bacterial isolates resistant to WATCH and RESEARVE antibiotics.**

| Strain type of isolate | n | Proportion of resistance isolates to watch and reserve antibiotics | | | |
|---|---|---|---|---|---|
| **Gram-positive (n=37)** | | Ofloxacin | Ceftriaxone | Vancomycin | Linezolid |
| *Staphylococcus aureus* | 18 | 3(16) | 5(27) | 6(33) | 5(27) |
| *Staphylococcus xylosus* | 18 | 4(22) | 6(33) | 5(26) | 4(21) |
| *Staphylococcus intermedius* | 1 | 0(0) | 1(100) | 0(0) | 0(0) |
| **Gram-positive (n=81)** | | | | | |
| *Escherichia coli* | 33 | 6(18) | 8(24) | NA | NA |
| *Enterobacter gergoviae* | 20 | 8(40) | 7(35) | NA | NA |
| *Sarratia liquefaciens* | 5 | 2(40) | 3(60) | NA | NA |
| *Serratia odorifera* | 4 | 1(25) | 1(25) | NA | NA |
| *Burkholderia pseudomallei* | 3 | 2(66) | 1(33) | NA | NA |
| *Pseudomonas aeruginosa* | 2 | 1(50) | 2(100) | NA | NA |
| *Aeromonas hydrophila 1* | 2 | 0(0) | 0(0) | NA | NA |
| *Klebsiella ornithinolytica* | 2 | 0(0) | 0(0) | NA | NA |
| *Yersinia enterocolitica* | 2 | 1(50) | 1(50) | NA | NA |
| *Enterobacter sakazakii* | 2 | 1(50) | 2(100) | NA | NA |
| *Enterobacter cloacae* | 2 | 2(100) | 1(50) | NA | NA |
| *Enterobacter aerogenes* | 1 | 0(0) | 0(0) | NA | NA |
| *Proteus vulgaris* | 1 | 0(0) | 0(0) | NA | NA |
| *Proteus mirabillis* | 1 | 0(0) | 1(100) | NA | NA |
| *Pausteurella haemolytica* | 1 | 0(0) | 0(0) | NA | NA |

NA: Not assessed in this study

### Vancomycin resistant genes

Considering that *Staphylococcus* species are frequently linked to wound infections, the presence of three genes associated with Vancomycin resistance were determined in this study. The results showed the presence of *Van*A in one *S. aureus* isolate (Fig 1), *Van*B in one isolate of *S. aureus* and *S. xylosus* (Fig 2), and *Van*C in four *S. aureus* and two *S. xylosus* isolates (Fig 3), respectively.

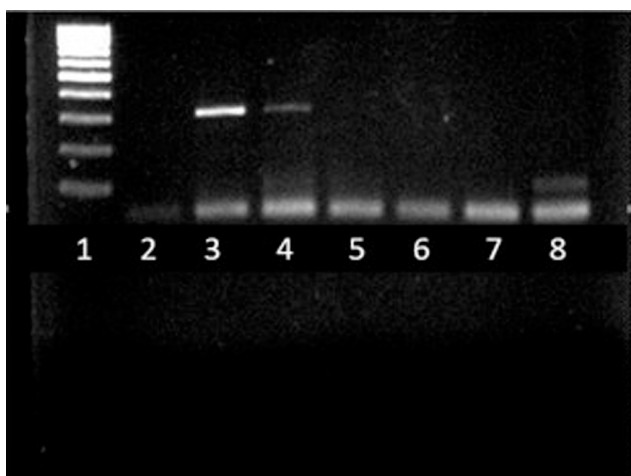

1. Molecular ladder (700 bp)
2. Negative control
3. Positive Control
4. Sample 1
5. Sample 2
6. Sample 3
7. Sample 4
8. Sample 5

**Fig 1. Electropherogram showing the detection of *Van*A gene in one *S. aureus* isolate.**

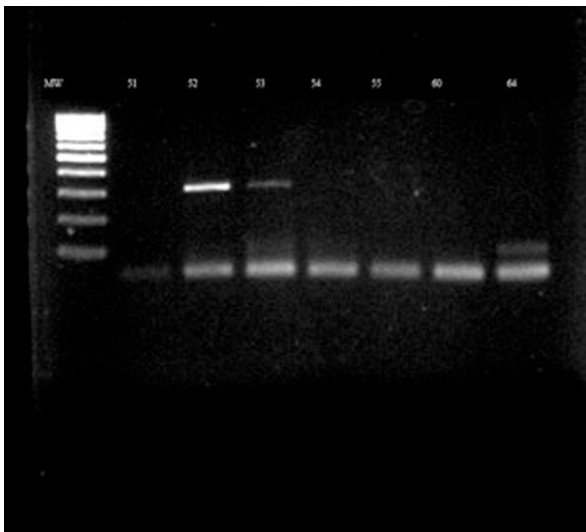

MW. Molecular ladder (5000 bp)
51. Negative control
52. Positive control
53. Sample 53
54. Sample 54
55. Sample 55
56. Sample 56
57. Sample 57
58. Sample 58
59. Sample 59
60. Sample 60
61. Sample 61

**Fig 2. Electropherogram showing the detection of *Van*B in two bacterial isolates (one *S. aureus* and one *S. xylosus*).**

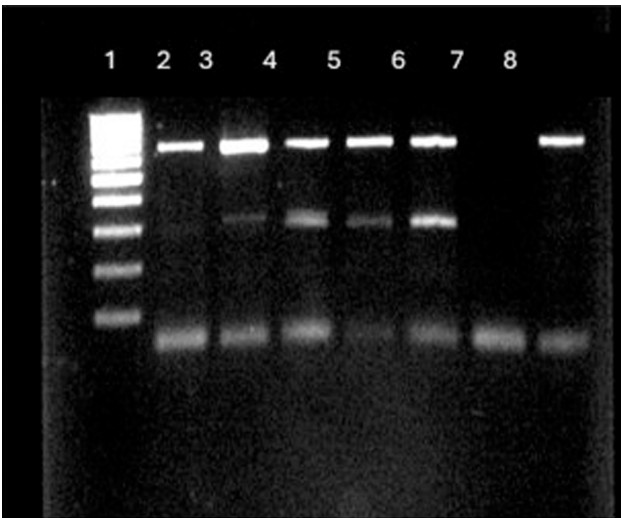

1. Molecular ladder (700 bp)
2. Positive control
3. Sample 1
4. Sample 2
5. Sample 3
6. Sample 4
7. Sample 5
8. Sample 6

**Fig 3. Electropherogram showing the detection of *Van* C in six isolates (four *S. aureus* and two *S. xylosus*).**

## Discussion

This study investigated the prevalence of various bacterial pathogens from patient wound infections, their resistance to commonly used antibiotics, and the presence of resistance genes, particularly to critical Watch and Reserve antibiotics at Kisii Teaching and Referral Hospital. By providing a comprehensive understanding of the local AMR landscape, this study sought to inform targeted interventions, improve empirical treatment protocols, and contribute to global efforts in combating antimicrobial resistance (AMR) [33,34].

The predominance of male patients and those aged 31–40 years with various wound types requiring clinical management observed in this study suggests occupational or lifestyle factors that increase exposure to wound-causing injuries,

highlighting the need for targeted prevention in high-risk groups [35]. Similar trends have been reported in studies from Ethiopia [19,36] and Nigeria [37], where occupational exposure was a significant risk factor for wound infections. A study at Jaramogi Oginga Odinga Teaching and Referral Hospital (JOOTRH) in Kenya also reported a higher prevalence of wound infections in male patients (59%) [30], further corroborating these findings. The high frequency of lower limb wounds compared to upper limb wounds aligns with findings from other regional studies, suggesting anatomical vulnerability and environmental exposure as critical factors in wound distribution [38]. The high occurrence of avulsion wounds may indicate the need for early intervention protocols tailored to mechanically induced injuries [38,39]. Furthermore, the relatively low incidence of burns underscores the likely influence of regional environmental and occupational conditions, offering valuable context for prioritising wound management strategies [40]. These insights into patient demographics and wound characteristics are vital for developing region-specific healthcare policies and interventions, especially in resource-limited settings [41].

The observed diversity of bacterial isolates reflects the complex microbial landscape of wound infections in hospital settings [5]. The high prevalence of Gram-negative bacteria, especially *E. coli* and *E. gergoviae* suggests a significant role for faecal contamination or environmental sources in these infections, warranting stricter hygiene and sanitation protocols [42]. Awuor et al. [30] found that *Klebsiella spp.* (14.8%) and *P. aeruginosa* (14.8%) were also dominant in chronic wound infections at JOOTRH, consistent with our findings. Similar findings have been reported in studies from Kenya [43], South Africa [44], Tanzania [45] and other African countries [46], where Gram-negative pathogens, particularly Enterobacteriaceae, were predominant in wound infections. The significant presence of Gram-positive bacteria like *S. aureus* and *S. xylosus* is consistent with studies from the United States of America [47], Saudi Arabia [48], and Poland [49], highlighting their role in wound pathophysiology and biofilm formation [50]. Besides, the low prevalence of rare isolates such as *P. mirabilis* and *P. haemolytica* highlights their opportunistic pathogenicity, often linked to immunocompromised hosts or specific environmental exposures [51]. These findings underline the importance of periodic surveillance of wound microbiota to capture emerging resistance patterns and inform empirical treatment choices. Additionally, recognising the unique microbial profiles in specific wound types may facilitate the development of targeted therapeutic and prophylactic interventions, ultimately improving patient outcomes.

The alarming resistance rates to ACCESS antibiotics observed in this study signify the limitations of first-line therapies and the escalating threat of antimicrobial resistance [52]. The near-universal resistance of *E. gergoviae* and *E. coli* to Cotrimoxazole-Trimethoprim and Ampicillin underscores the urgency of revising empirical treatment protocols [53]. Comparable resistance patterns have been documented in Nigeria [54] and Ethiopia [55], where widespread resistance to first-line antibiotics has rendered them ineffective in treating wound infections. Moreover, a similar study in the Nyanza region of Kenya reported high resistance rates to Cotrimoxazole (48.1%), Clindamycin (25.9%), and Erythromycin (25.9%) among wound isolates [30]. Resistance to Nitrofurantoin, Streptomycin, and Nalidixic Acid among other antibiotics further compounds the therapeutic challenge, especially for community-acquired infections [56]. Such widespread resistance necessitates urgent investment in antimicrobial stewardship programmes to preserve the efficacy of existing antibiotics [57].

The moderate resistance to Tetracycline and Gentamicin, though relatively lower, underscores their potential utility in some cases while highlighting the need for continuous monitoring to prevent escalation [58]. A previous report from Ghana [59] showed moderate resistance to Tetracycline and Gentamicin, suggesting their continued but cautious utility in treatment regimens. Integrating local resistance data into clinical guidelines will optimise treatment regimens and minimise therapeutic failures in regions with similar microbial profiles [60]. Furthermore, the observed resistance patterns call for the adoption of precision medicine approaches that include pathogen-specific susceptibility testing to guide effective treatment [61].

The emergence of resistance to Watch and Reserve antibiotics, including Ofloxacin and Ceftriaxone, among both Gram-positive and Gram-negative isolates, is particularly concerning [52]. These antibiotics, often regarded as last-resort

treatments, are critical for managing multidrug-resistant infections [62]. Resistance to Vancomycin and Linezolid among Gram-positive isolates, including *S. aureus*, has been similarly reported in hospital settings in China [63] and Egypt [64], raising concerns about limited treatment options for severe infections, especially in resource-constrained settings where alternative treatments may be limited [61,65]. The high resistance observed in *P. aeruginosa* and *E. sakazakii* to Oflox-acin and Ceftriaxone aligns with findings from studies in Brazil [66] among other countries [67], where these pathogens demonstrated resistance trends exacerbated by hospital-acquired infections, underscoring the need for strict antibiotic use policies to limit the emergence of resistance in nosocomial settings [62]. These findings further highlight the importance of infection prevention and control measures to reduce the spread of resistant strains [68]. The apparent cross-resistance among critical antibiotics raises the spectre of therapeutic dead-ends, necessitating policy-driven interventions that priori-tise antibiotic conservation and effective stewardship programmes at all levels of care [69].

The molecular detection of Vancomycin resistance genes (*Van*A, *Van*B, *Van*C) in *S. aureus* and *S. xylosus* provides a crucial link between genotypic and phenotypic resistance [70]. The presence of these genes suggests horizontal gene transfer as a potential driver of resistance, raising concerns about the genetic adaptability of bacterial pathogens [71]. Studies from Egypt [72] have also reported the presence of Vancomycin resistance genes in clinical isolates, emphasizing the need for molecular surveillance in AMR monitoring programs [61]. Locally, Awuor et al. [30] identified virulence fac-tors such as hemolysin and protease production in 75% of isolates, further complicating treatment outcomes. Identifying resistance genes also offers opportunities for the development of rapid diagnostic tools to detect resistant pathogens and implement timely interventions [65]. Additionally, understanding the genetic mechanisms underpinning resistance could inform the discovery of novel therapeutic targets, thereby bolstering efforts to counteract the AMR crisis.

From a research perspective, these findings underscore the importance of prioritising the development of novel anti-microbial agents and exploring alternative therapeutic approaches. The high prevalence of multidrug-resistant patho-gens calls for innovative strategies such as combination therapies, bacteriophage applications, and immunotherapies [62]. Future studies should also investigate the role of local environmental and healthcare practices in driving AMR, as such insights are critical for designing tailored interventions. Understanding these contextual factors will help in design-ing interventions that are both effective and sustainable in diverse clinical settings. The integration of advanced omics technologies into routine diagnostics may further refine our understanding of resistance mechanisms and pathogen-host interactions, thereby advancing personalised medicine in infectious disease management.

This study highlights the need to strengthen antimicrobial stewardship and infection control policies, especially in tertiary healthcare institutions like Kisii Teaching and Referral Hospital. Implementing national surveillance systems to capture resistance patterns will support evidence-based decision-making and resource allocation [73]. Public awareness campaigns on the judicious use of antibiotics can complement these measures, reducing the demand for inappropriate prescriptions [60]. Additionally, fostering collaborations between healthcare facilities, researchers, and policymakers can help address the multifaceted challenges posed by AMR. Strengthening laboratory capacity to routinely conduct suscepti-bility testing and monitor resistance trends is essential for effective policy implementation and sustained AMR mitigation.

## Limitations

This study has several limitations that warrant consideration. Firstly, the sample size, though sufficient for initial insights, may not fully capture the microbial diversity and resistance patterns in other healthcare settings or broader populations. Secondly, the study was conducted in a single tertiary hospital, potentially limiting the generalisability of findings to other regions with differing healthcare practices and environmental factors. Thirdly, the reliance on phenotypic methods for anti-microbial susceptibility testing, while robust, may not detect all resistance mechanisms, underscoring the need for comple-mentary molecular analyses. Lastly, the cross-sectional design precludes the assessment of temporal trends in resistance patterns, which could provide a more dynamic understanding of antimicrobial resistance. Future research should address these limitations through multicentre studies, larger sample sizes, and longitudinal designs.

## Conclusions and recommendations

This study showed the presence of a wide variety of bacterial strain strains in infected wounds of patients at Kisii Teaching and Referral Hospital. Besides the findings underscore the significant burden of multidrug-resistant pathogens in wound infections, demonstrated by high resistance rates to both ACCESS antibiotics, such as Cotrimoxazole-Trimethoprim (97%) and Ampicillin (92%), and critical Watch and Reserve antibiotics, including Vancomycin (29%) and Ceftriaxone (33%). The detection of resistance genes like *VanA, VanB,* and *VanC* in *Staphylococcus* species highlights the genetic mechanisms underpinning resistance and the potential for horizontal gene transfer. Socio-demographic and clinical factors, such as the predominance of male patients, lower limb wounds, and avulsions, may further inform targeted prevention strategies. These findings emphasise the urgent need for evidence-based antimicrobial stewardship to mitigate therapeutic failures and protect patient outcomes in high-burden settings.

Strengthening infection prevention and control measures, such as standardised wound care protocols and enhanced hygiene practices, is critical to curbing the spread of resistant pathogens. Policymakers should prioritise the establishment of robust AMR surveillance systems and the integration of local resistance data into clinical guidelines to optimise empirical therapy. Public education campaigns to promote the rational use of antibiotics can complement these measures. Future research should focus on developing alternative therapeutic strategies, such as combination therapies or bacteriophage applications, and investigating the environmental and healthcare-related drivers of AMR to inform sustainable interventions. These efforts can significantly contribute to combating AMR and improving patient outcomes in resource-constrained healthcare settings.

## Supporting information

**S1_File. Source file (Microsoft Excel Workbook) containing the raw data collected in this study.**
(PDF)

## Acknowledgments

We are grateful for the technical assistance provided by Mr. Kiyondi, Mr. D. Ounga and Mrs. R. Anyango of Kisii Teaching, and Referral Hospital (Microbiology laboratory).

## Author contributions

**Conceptualization:** Samson Onsando, Wycliffe Masanta, Andrew Nyerere, Moses Njire.

**Data curation:** Samson Onsando, Wycliffe Masanta, Andrew Nyerere, Moses Njire, Gervason Moriasi.

**Formal analysis:** Samson Onsando.

**Funding acquisition:** Samson Onsando.

**Investigation:** Samson Onsando, Gervason Moriasi.

**Methodology:** Samson Onsando, Andrew Nyerere, Gervason Moriasi.

**Project administration:** Samson Onsando.

**Software:** Gervason Moriasi.

**Supervision:** Wycliffe Masanta, Andrew Nyerere, Moses Njire, Gervason Moriasi.

**Validation:** Samson Onsando, Wycliffe Masanta, Andrew Nyerere, Moses Njire, Gervason Moriasi.

**Visualization:** Samson Onsando, Wycliffe Masanta, Andrew Nyerere, Moses Njire, Gervason Moriasi.

**Writing – original draft:** Samson Onsando, Wycliffe Masanta, Andrew Nyerere, Moses Njire, Gervason Moriasi.

**Writing – review & editing:** Samson Onsando, Wycliffe Masanta, Andrew Nyerere, Moses Njire, Gervason Moriasi.

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
