## [Decision Letter · Decision Letter 0]

Apr 03 2025

Dear Dr. Onsando,

Thank you for submitting your manuscript to PLOS ONE. After careful consideration, we feel that it has merit but does not fully meet PLOS ONE’s publication criteria as it currently stands. Therefore, we invite you to submit a revised version of the manuscript that addresses the points raised during the review process.

We look forward to receiving your revised manuscript.

Kind regards,

Mabel Kamweli Aworh, DVM, MPH, PhD. FCVSN

Academic Editor

PLOS ONE

3. Please include a separate caption for each figure in your manuscri

Additional Editor Comments:

In additional to addressing all the reviewers' comments, please update your reference list with studies done within the last five years. Old references should account for 20% of the list and not more, please.

Reviewers' comments:

Reviewer's Responses to Questions

**Comments to the Author**

1. Is the manuscript technically sound, and do the data support the conclusions?

Reviewer #1: Yes

Reviewer #2: Yes

2. Has the statistical analysis been performed appropriately and rigorously?

Reviewer #1: Yes

Reviewer #2: Yes

3. Have the authors made all data underlying the findings in their manuscript fully available?

Reviewer #1: Yes

Reviewer #2: Yes

4. Is the manuscript presented in an intelligible fashion and written in standard English?

Reviewer #1: Yes

Reviewer #2: Yes

Reviewer #1: The authors have done a great job with this research by carefully explaining their methods and procedure for sample collection. Also worthy of note is the discussion involves a good description of possible explanations and mechanisms involved for the wounds as well as the diversity and antibiotic susceptibility seen in the results

Here are a few comments (remaining are annotated in the attached file) and clarifications to improve the readability and context of the manuscript

ABSTRACT

1. Please include the frequency (count) together with each percentage for better understanding of the undelying data

METHODS

1. There needs to be a description of how the data was recorded. Was any tool used? what type of tool/instrument was used to collect data from the 149 patients?? (electronic database, standardized forms e.t.c) Please include the description (including the type, structure, form, pretesting e.t.c) and state the validity statistics of the tool used.

2. In the description of the STUDY SITE, It is important to include descriptions of factors in the study site to help improve the context of your study especially as it relates to wounds and microbial infection

Please include information concerning these issues

- Why this study site was chosen - Explain why studying wound infections specifically at the Surgical Unit of this Hospital is important. Also it is important to mention any unique aspects of this setting or patient population that make it a valuable focus for your research.

- What is the Laboratory capacity of the study site? This is clearly missing

- Include the Actual location of the site (latitude and longitude)

- What are the different types of wounds commonly seen in the study area

- Common sources of wounds i.e activities/diseases that leads to common wounds in the study site

- The average number of patients seen during the study period (study population) this is important for sample size determination!

- what is the availability of an Antimicrobial stewardship program and IPC policy in the hospital?

3. For the Study Population and sample size - It is important that you properly define your inclusion criteria. what you said below should be reconciled with this. what is stated in the sentence above is not very clear to me. Are you including ANY patient with a wound that shows ANY clinical sign of wound infection or MORE THAN ONE sign of wound infection? Please make this clear and include it in the selection criteria section

4. For the SAMPLE SIZE CALCULATIONS- This is not the correct formula for sample size calculation for your study design (cross-sectional study). This formular is used only when you have 2 groups like in a case control study or experimental study. Please kindly look for the correct formula to use and recalculate the minimum sample size. This is inaccurate!!!

5. Also, in the sample size computation, Can you indicate why you decided to correct for finite population?

Also when you do a finite population correction, the calculated sample size reduces not increases

Please explain in detail how you computed finite population correction.

6. For the sub-section for Inclusion and exclusion criteria - How did you determine patients with "clinically infected wounds"? It is important to this reconcile with your statement above in the study population section

7. Under the data analysis section - It is unclear how the data was entered into GraphPad software, please explain more

8. can you include the standard reference for this software (Graphpad) and add to the reference list?

RESULTS

1. It is important to begin the results section by stating the total number of patients that were approached, the number that agreed to the study and the response rate (number of completed forms divided by total number of patients that were approached . Please included that in your results

2. percentages and proportions dont mean much without frequencies and totals. Please include the total number of patients in this section, and the frequencies for each percentage. it is always important to include frequencies together with their respective percentages everytime you analyse categorical variables

3. Table 3: what other clinical characteristics of the patients did you collect?

why is there no information about the Medical history of the patient (e.g comorbidities, previous surgeries). This is important for wound healing and infection. what about disease-specific factors ie wound type, presence of clinical signs of wound infections, including purulent discharge, erythema, swelling, or delayed wound healing. These are what you mentioned when describing your inclusion criteria

4. Table 3: This is incorrect! Please remove, a total is not needed here. If all the variables had the same number of respondents (ie no missing values) then include the number of respondents ie n=149 either in the title or the frequency column header.

5. For all tables in the results section, include the number of respondents in all the titles or frequency columns

DISCUSSION

1. it is important that you compare your findings with similar studies locally, regionally, and globally. Please do this for your major objectives like

- Diversity of Bacterial Isolates

- Antibiotic Susceptibility Profiles of Bacterial Isolates.

2. In your description of the study site, you did not mention that the hospital had AMS and IPC policies. It is important that this is stated.

3. Limitations: unfortunayely, the sample size was not correctly calculated. please revise.

Reviewer #2: Dear Authors,

It is obvious that you put a lot of resources into this work and the outcome is commendable.

I have no recommendations for a revision at the moment and wish you the very best in your future publications.

**Do you want your identity to be public for this peer review?** For information about this choice, including consent withdrawal, please see our Privacy Policy

Reviewer #1: **Yes: ** Abdulhakeem Abayomi Olorukooba

Reviewer #2: **Yes: ** Dr. Oluwafolayemi Doyeni

---

## [Author Response · Author response to Decision Letter 1]

11 Mar 2025

PLOS ONE

Responses to Review Comments

Manuscript Title: Diversity and antibiotic susceptibility profiles of bacterial isolates from wound infections in patients at the surgical unit of Kisii Teaching and Referral Hospital, Kenya

Manuscript ID: PONE-D-25-04326

Dear Editors and Reviewers,

I heartly thank you for taking your valuable time to review our manuscript. Moreover, I appreciate your positive criticism, strong comments, and suggestions to improve the quality of our research article. We have carefully studied all the review comments and revised our manuscript accordingly. Evidently, the quality of our revised research article has improved significantly based on your critique and helpful suggestions. It is our hope that our revised manuscript will meet the standards and approval criteria for publication in this esteemed Journal of PLoS ONE. Please find our point-by-point responses to the Editor and reviewer comments (highlighted yellow) and the author responses (indicated in green) below. The changes made to the revised manuscript are highlighted green for your reference.

Editor

1. In additional to addressing all the reviewers' comments, please update your reference list with studies done within the last five years. Old references should account for 20% of the list and not more, please.

Authors’ response: Dear Editor, thank you for this suggestion. We have revised all the references, ensured they are current and fall within five years, and only retained a few (<5%) where necessary. Additionally, we have reviewed the author guidelines and aligned our manuscript accordingly.

Authors’ response: We have provided a complete data availability statement in the revised manuscript under declarations and the online submission form as advised.

3. Please include a separate caption for each figure in your manuscript

Authors’ response: We have separated the figures and included separate captions for each figure as recommended.

Reviewer 1

1. ABSTRACT

Please include the frequency (count) together with each percentage for better understanding of the underlying data.

Authors’ response: Thank you for this suggestion. Accordingly, we have included the frequencies along with their percentages in the abstract section of the revised manuscript.

METHODS

1. There needs to be a description of how the data was recorded. Was any tool used? what type of tool/instrument was used to collect data from the 149 patients?? (electronic database, standardized forms e.t.c) Please include the description (including the type, structure, form, pretesting e.t.c) and state the validity statistics of the tool used.

Authors’ response: We appreciate your keen review. Notably, we did not use a questionnaire or checklist to collect patient information in this study. The demographic data (patient gender and age) and clinical information (Type of wound and location of the wound) were collected using a standard clinical form designed and approved by the Kisii Teaching and Referral Hospital, and have included this information in the revised manuscript

2. In the description of the STUDY SITE, It is important to include descriptions of factors in the study site to help improve the context of your study especially as it relates to wounds and microbial infection

Please include information concerning these issues

- Why this study site was chosen - Explain why studying wound infections specifically at the Surgical Unit of this Hospital is important. Also it is important to mention any unique aspects of this setting or patient population that make it a valuable focus for your research.

- What is the Laboratory capacity of the study site? This is clearly missing

- Include the Actual location of the site (latitude and longitude)

- What are the different types of wounds commonly seen in the study area

- Common sources of wounds i.e activities/diseases that leads to common wounds in the study site

- The average number of patients seen during the study period (study population) this is important for sample size determination!

- what is the availability of an Antimicrobial stewardship program and IPC policy in the hospital?

Authors’ response: We have considered your comments and rewritten the description of the study site and added the suggested information, including the location, hospital capacity, average number of patients served per month, wound types managed, local setting, and the availability of antimicrobial stewardship program and IPC policy in KUTRH. Thank you for these observations.

3. For the Study Population and sample size - It is important that you properly define your inclusion criteria. what you said below should be reconciled with this. what is stated in the sentence above is not very clear to me. Are you including ANY patient with a wound that shows ANY clinical sign of wound infection or MORE THAN ONE sign of wound infection? Please make this clear and include it in the selection criteria section.

Authors’ response: Thank you for your comment and apologies for the confusion. Accordingly, we have revised the inclusion criteria and emphasized that patients presenting with clinically diagnosed infected surgical wounds, as confirmed by clinical evaluation (e.g., presence of purulent discharge, erythema, warmth, swelling, or pain), had not received any antibiotic treatment within the preceding 48 hours, and provided informed consent to participate were included in this study.

4. For the SAMPLE SIZE CALCULATIONS- This is not the correct formula for sample size calculation for your study design (cross-sectional study). This formular is used only when you have 2 groups like in a case control study or experimental study. Please kindly look for the correct formula to use and recalculate the minimum sample size. This is inaccurate!!!.

Authors’ response: Thank you so much for this observation and receive sincere apologies for this mistake. We have revised the study population and sample calculation formular and highlighted how the sample size was arrived at in the revised manuscript. In brief, 221 patients were the anticipated sample size; however, due to various limitations such as lack of consent from some patients, the response rate was 67% (149 patients), and was considered statistically significant for detecting trends in bacterial diversity, susceptibility, and resistance patterns, given the prevalence of wound infections reported in a similar study setting.

5. Also, in the sample size computation, Can you indicate why you decided to correct for finite population?

Also when you do a finite population correction, the calculated sample size reduces not increases

Please explain in detail how you computed finite population correction.

Authors’ response: We have carefully revised the sample calculation and detailed the rationale for the finite population correction, in a stepwise manner for clarity. Thank you for this helpful suggestion.

6. For the sub-section for Inclusion and exclusion criteria - How did you determine patients with "clinically infected wounds"? It is important to this reconcile with your statement above in the study population section

Authors’ response: We have clarified the inclusion and exclusion criteria and explained how infected wounds were determined in eligible patients.

7. Under the data analysis section - It is unclear how the data was entered into GraphPad software, please explain more.

Authors’ response: Based on your suggestion, we have clarified how the data were entered into GraphPad Prism software and included details of descriptive statistics performed.

8. can you include the standard reference for this software (Graphpad) and add to the reference list?

Authors’ response: Thank you for this recommendation. Accordingly, we have included a citation and reference for the GraphPad Prism software.

RESULTS

1. It is important to begin the results section by stating the total number of patients that were approached, the number that agreed to the study and the response rate (number of completed forms divided by total number of patients that were approached. Please included that in your results.

Authors’ response: We appreciate you for this interesting insight. We have included the suggested details under Demographic and clinical characteristics of study patients in the results section of the revised manuscript.

2. percentages and proportions dont mean much without frequencies and totals. Please include the total number of patients in this section, and the frequencies for each percentage. it is always important to include frequencies together with their respective percentages everytime you analyse categorical variables

Authors’ response: Thank you so much for this recommendation. We have included the actual frequencies together with the respective percentages in all descriptions suggested details under the results sections. We agree that this format clarifies our results and makes it easy to infer for readers.

3. Table 3: what other clinical characteristics of the patients did you collect?

why is there no information about the Medical history of the patient (e.g comorbidities, previous surgeries). This is important for wound healing and infection. what about disease-specific factors ie wound type, presence of clinical signs of wound infections, including purulent discharge, erythema, swelling, or delayed wound healing. These are what you mentioned when describing your inclusion criteria.

Authors’ response: We acknowledge your concern, especially on the aspect of wound healing. Although factors such as comorbidities and previous surgeries may influence would healing, this was not our main focus in this study. Our interest was infected wounds-based on clinical evaluation, regardless of comorbidities, the bacterial strains they harboured, and their antimicrobial susceptibility profile. It is our view that the additional clinical factors maybe be well aligned with studies investing immunity and wound healing and will certainly incorporate these in our future expanded studies.

4. Table 3: This is incorrect! Please remove, a total is not needed here. If all the variables had the same number of respondents (ie no missing values) then include the number of respondents ie n=149 either in the title or the frequency column header.

Authors’ response: Thank you so much for your suggestion. Accordingly, we have removed the number of patients and instead included the totals on the table heading.

5. For all tables in the results section, include the number of respondents in all the titles or frequency columns.

Authors’ response: Thank you. Based on this advice, we have included the totals in the table/column headings in the revised manuscript.

DISCUSSION

1. it is important that you compare your findings with similar studies locally, regionally, and globally. Please do this for your major objectives like

- Diversity of Bacterial Isolates

- Antibiotic Susceptibility Profiles of Bacterial Isolates.

Authors’ response: We appreciate your keen review and recommendations to improve and enrich the quality of our manuscript’s discussion. We have considered your insights and revised the discussion section and incorporated findings from other similar studies conducted locally and other countries and highlighted the implications and recommendations appropriately.

2. In your description of the study site, you did not mention that the hospital had AMS and IPC policies. It is important that this is stated.

Authors’ response: Thank you for this suggestion. We have included this information in the methodology section under the description of the study site.

3. Limitations: unfortunately, the sample size was not correctly calculated. please revise.

Authors’ response: Thank you for this comment. We have revised the sample size calculation formular and procedure and hope this addresses the raised concern.

Reviewer 2

Dear Authors,

It is obvious that you put a lot of resources into this work and the outcome is commendable.

I have no recommendations for a revision at the moment and wish you the very best in your future publications.

Authors’ response: We thank you for taking time to review our manuscript and your encouraging comment.

Thank you so much for helping us to improve the quality our research article. We look forward to your feedback.

---

## [Decision Letter · Decision Letter 1]

May 17 2025

Thank you for submitting your manuscript to PLOS ONE. After careful consideration, we feel that it has merit but does not fully meet PLOS ONE’s publication criteria as it currently stands. Therefore, we invite you to submit a revised version of the manuscript that addresses the points raised during the review process.

We look forward to receiving your revised manuscript.

Kind regards,

Mabel Kamweli Aworh, DVM, MPH, PhD. FCVSN

Academic Editor

PLOS ONE

Journal Requirements:

Reviewers' comments:

Reviewer's Responses to Questions

**Comments to the Author**

Reviewer #1: (No Response)

Reviewer #2: All comments have been addressed

2. Is the manuscript technically sound, and do the data support the conclusions?

Reviewer #1: Yes

Reviewer #2: (No Response)

3. Has the statistical analysis been performed appropriately and rigorously?

Reviewer #1: Yes

Reviewer #2: (No Response)

4. Have the authors made all data underlying the findings in their manuscript fully available?

Reviewer #1: Yes

Reviewer #2: (No Response)

5. Is the manuscript presented in an intelligible fashion and written in standard English?

Reviewer #1: Yes

Reviewer #2: (No Response)

Reviewer #1: The authors have made tremendous improvements in the quality of the manuscript. I must commend all their efforts. However, there is still an issue with the sample size calculation which unfortunately still needs to be addressed. The formula is now correct however the prevalence (p) used in the formula should be the prevalence of Antimicrobial resistance among patients with wound infection rather than the prevalence of wound infection. Since you’re exclusively studying patients who already have wound infections, Your sample size calculation should then focus on estimating antimicrobial resistance within that population of patients with wound infection. In this scenario, you should use the prevalence of antimicrobial resistance among patients with wound infections from prior studies, since resistance is a key outcome.

Similarly in the inclusion and exclusion criteria, there is no need to mention "consent" as a criteria for enrolment into a study. This is obvious. Its unethical to even enroll anyone in a research without consent

I noticed you have some minors as study subjects. It is important to include how you obtained "assent" from these participants and consent from their caregivers.

Reviewer #2: (No Response)

**Do you want your identity to be public for this peer review?** For information about this choice, including consent withdrawal, please see our Privacy Policy

Reviewer #1: **Yes: ** Abdulhakeem Abayomi Olorukooba

Reviewer #2: **Yes: ** Dr. Oluwafolayemi Doyeni

---

## [Author Response · Author response to Decision Letter 2]

12 Apr 2025

Dear Editors and Reviewers,

I heartly thank you for taking your valuable time to review our manuscript. Moreover, I appreciate your positive criticism, strong comments, and suggestions to improve the quality of our research article. We have carefully studied all the review comments and revised our manuscript accordingly. Evidently, the quality of our revised research article has improved significantly based on your critique and helpful suggestions. It is our hope that our revised manuscript now meets the standards and approval criteria for publication in this esteemed Journal of PLoS ONE, having addressed all the concerns. Please find our point-by-point responses to the reviewer’s comments (highlighted yellow) and the author responses (indicated in green) below.

Reviewer 1

1. The authors have made tremendous improvements in the quality of the manuscript. I must commend all their efforts. However, there is still an issue with the sample size calculation which unfortunately still needs to be addressed. The formula is now correct however the prevalence (p) used in the formula should be the prevalence of Antimicrobial resistance among patients with wound infection rather than the prevalence of wound infection. Since you’re exclusively studying patients who already have wound infections, Your sample size calculation should then focus on estimating antimicrobial resistance within that population of patients with wound infection. In this scenario, you should use the prevalence of antimicrobial resistance among patients with wound infections from prior studies, since resistance is a key outcome. Authors’ response: Thank you so much for your keen review and helpful suggestion. We have now amended the prevalence to that of antimicrobial resistance among patients with wound infections and recalculated the sample size accordingly.

2. Similarly in the inclusion and exclusion criteria, there is no need to mention "consent" as a criteria for enrolment into a study. This is obvious. Its unethical to even enroll anyone in a research without consent’ Authors’ response: Thank you for this suggestion. Accordingly, we have expunged the last sentence in this section as it is unnecessary.

3. I noticed you have some minors as study subjects. It is important to include how you obtained "assent" from these participants and consent from their caregivers. Authors’ response: Thank you for observation. For patients aged below 18 years, their parents or legal guardians provided informed consent according to ethical research guidelines.

Thank you so much for helping us to improve the quality our research article. We look forward to your feedback.

---

## [Decision Letter · Decision Letter 2]

Diversity and antibiotic susceptibility profiles of bacterial isolates from wound infections in patients at the surgical unit of Kisii Teaching and Referral Hospital, Kenya

PONE-D-25-04326R2

Dear Dr. Onsando,

We’re pleased to inform you that your manuscript has been judged scientifically suitable for publication and will be formally accepted for publication once it meets all outstanding technical requirements.

Kind regards,

Mabel Kamweli Aworh, DVM, MPH, PhD. FCVSN

Academic Editor

PLOS ONE

Additional Editor Comments (optional):

Reviewers' comments:

Reviewer's Responses to Questions

**Comments to the Author**

Reviewer #1: All comments have been addressed

Reviewer #2: All comments have been addressed

2. Is the manuscript technically sound, and do the data support the conclusions?

Reviewer #1: Yes

Reviewer #2: (No Response)

3. Has the statistical analysis been performed appropriately and rigorously?

Reviewer #1: Yes

Reviewer #2: (No Response)

4. Have the authors made all data underlying the findings in their manuscript fully available?

Reviewer #1: Yes

Reviewer #2: (No Response)

5. Is the manuscript presented in an intelligible fashion and written in standard English?

Reviewer #1: Yes

Reviewer #2: (No Response)

Reviewer #1: Authors have successfully addressed all my concerns. I dont have any more comments or questions. Thank you

Reviewer #2: (No Response)

**Do you want your identity to be public for this peer review?** For information about this choice, including consent withdrawal, please see our Privacy Policy

Reviewer #1: **Yes: ** ABDULHAKEEM ABAYOMI OLORUKOOBA

Reviewer #2: **Yes: ** Dr. Oluwafolayemi Doyeni

---

## [Editor Report · Acceptance letter]

PONE-D-25-04326R2

PLOS ONE

Dear Dr. Onsando,

I'm pleased to inform you that your manuscript has been deemed suitable for publication in PLOS ONE. Congratulations! Your manuscript is now being handed over to our production team.

Kind regards,

on behalf of

Dr. Mabel Kamweli Aworh

Academic Editor

PLOS ONE